# Age Matters: Key Contributors to Interferon Toxicity in Infants During Influenza Virus Infection

**DOI:** 10.3390/v17071002

**Published:** 2025-07-17

**Authors:** Abigail P. Onufer, Alison J. Carey

**Affiliations:** 1Department of Microbiology and Immunology, Drexel University College of Medicine, Philadelphia, PA 19102, USA; apo38@drexel.edu; 2Department of Pediatrics, St. Christopher’s Hospital for Children, Drexel University College of Medicine, Philadelphia, PA 19134, USA

**Keywords:** interferon (IFN), influenza virus (IV), lung, neonate, repair, barrier, reactive oxygen species (ROS)

## Abstract

Respiratory viral infections are a leading cause of early childhood hospitalizations in the United States. Neonatal immune responses are reliant on innate mechanisms during the first few months of life. Interferons (IFNs) are a key component of this response. These antiviral cytokines are produced early in infection and aid in viral control and clearance. Although generally considered protective in the setting of respiratory viral infections, the recent literature has suggested that IFNs may exacerbate disease. In the process of promoting an antiviral environment, IFNs impede cell proliferation, contribute to pulmonary barrier disruption, and generate reactive oxygen species. This is not tolerated in the rapidly developing neonatal lung. Therefore, IFNs contribute to pathogenesis in the influenza-infected neonate. This review focuses on the potential mechanisms that drive IFN-induced toxicity in neonates and prospective therapeutics to mitigate this toxicity.

## 1. Introduction

Influenza virus (IV) infections are a global health burden. In the United States, IV infections account for 120,000–710,000 hospitalizations and 6,300–52,000 deaths annually [1]. Prematurity is the leading risk factor for increased susceptibility and morbidity to respiratory viral infections [2]. In fact, premature infants display the highest hospitalization and intubation rates in the pediatric population [3,4]. Immunological developmental differences are hypothesized to be the drivers of severe disease in this population. Influenza vaccines are available and recommended for children above the age of 6 months; however, infants under 6 months, the age group most at risk, are not eligible. Therefore, investigation of age-specific immune responses that contribute to infant morbidity is crucial to developing effective therapeutics.

Interferons (IFNs) are a family of cytokines that drive initial antiviral responses following infection [5]. IFNs also play important roles in tissue homeostasis, cell signaling, and immune regulation. IFNs are divided into three classes: type I IFNs (IFN-Is; αβ), type II IFNs (IFN-IIs; γ), and type III IFNs (IFN-IIIs; λ), based on their sequence homology and receptor affinity [6]. IFN-Is and IFN-IIIs are the primary IFNs responsible for antiviral protection in the lung [5,6,7,8,9]. Host pattern recognition receptors in the lungs, such as toll-like receptors, RIG-I-like receptors, NOD-like receptors, and Z-DNA binding protein, recognize and respond to viral pathogen-associated molecular patterns, including viral proteins and RNA [10,11,12]. IFN-Is and IFN-IIIs release in response to viral recognition and signal in an autocrine or paracrine manner within the infected respiratory tract [7,13].

Both IFN-Is and IFN-IIIs activate JAK-STAT to induce an overall antiviral state in the infected tissue through the expression of IFN-stimulated genes (ISGs), which impair viral replication and protein synthesis, and direct the immune response against the virus [6]. Some have reported how IFN-Is and IFN-IIIs can induce overlapping expressions of ISGs in primary murine tracheal epithelial cells infected with IV [14]. However, Zhou et al. reported a unique subset of ISGs induced by IFN-I stimulation, but not by IFN-III, using a human lymphocyte cell line [15]. Similarly, Davidson et al. demonstrated that IFN-III treatment induced redundant ISG expression in the whole lung relative to IFN-I treatment, while IFN-I treatment uniquely upregulated genes associated with cytokine production and immune cell recruitment in a murine model of IV infection [16]. This data suggests that IFN-IIIs only induce a subset of total ISGs produced by IFN-I signaling. Furthermore, IFN-I signaling generates a larger magnitude of ISG expression relative to IFN-III signaling [15]. In addition, IFN-Is drive rapid, transient ISG expression, while IFN-IIIs have a delayed induction of ISGs but sustained expression [17,18].

Differences in the cellular expression of IFN-I and IFN-III receptors contribute to the variations in ISG kinetics and magnitude. IFN-Is are produced and recognized by all nucleated cells due to the ubiquitous expression of the IFN-αβ receptor 1 (IFNAR1) and IFNAR2 receptors [6]. Therefore, IFN-Is induce a robust and systemic response during viral infections. IFN-I signaling has been linked to excessive inflammation and impaired lung repair in adult murine models of IV infection [19,20,21]. Davidson et al. reported that IFN-I stimulation, but not IFN-III, led to an increase in inflammatory cytokine production, such as IL-6 and CCL2, from human peripheral blood mononuclear cells (PBMCs) [16]. However, the disruption of IFN-I signaling leads to impaired viral clearance and enhanced morbidity [22,23]. Unlike IFN-Is, IFN-IIIs signal through IFNLR1 and IL-10Rβ, the former of which is expressed primarily on epithelial cells and some immune cells at mucosal sites [7,24]. The IFN-III response is less potent and more localized, promoting immunity at barrier sites like the lung [6,8,16]. IFN-IIIs have been identified as the primary IFNs produced in the airway following IV infection, and adult mice lacking IFN-III receptor signaling exhibit increased viral loads [25,26]. It is hypothesized that IFN-III signaling functions to induce antiviral immunity without the systemic inflammation and bystander damage initiated by IFN-I signaling [16,17]. However, IFN-III signaling also contributes to cell death and barrier disruption in the infected adult lung [21,27,28].

Using a neonatal murine model of IV infection, we have recently shown that IFN-Is are the main driver of pathogenesis, while IFN-IIIs serve a primarily protective role [23]. The protective effect of IFN-IIIs in the neonatal lung have also been illustrated during murine pneumovirus infection [29]. This suggests age-specific differences in the role of IFNs in infected neonates. Understanding the relative contribution of IFN-Is and -IIIs to IV antiviral protection or pathogenesis has the potential to inform therapeutics that will specifically target this high-risk age group.

## 2. Age-Specific Response to Respiratory Viral Infections

### 2.1. Tolerance Drives Dampened Neonatal Immune Responses

At birth, neonates are introduced to an oxygen-rich environment filled with both commensals and pathogens. During this time, immunological tolerance drives the balance between commensal colonization and pathogen clearance [30]. As a result, tolerance contributes to impaired T cell functionality and less diverse TCR repertoires in the neonate [31,32,33,34]. Neonates are more reliant on innate immune mechanisms. However, these innate defenses are also not fully developed. Studies using infant PBMCs or cord blood mononuclear cells (CBMCs) have shown defects in innate responses such as decreased macrophage effector capabilities and increased expression of inhibitory receptors [35]. Therefore, the neonatal response to pathogens is not as robust as what is observed in adults (Figure 1).

To address this unique developmental period, our group uses a clinically relevant neonatal model of intranasal IV infection. Neonatal animal models have become a useful surrogate to study age-specific responses to various infections, including *Bordetella pertussis*, respiratory syncytial virus (RSV), and severe acute respiratory syndrome coronavirus 2 (SARS-CoV-2) [38,39,40]. Similarly to a human infant, neonatal mice undergo extensive lung development following birth and exhibit dampened T cell responses to pathogens [31,41]. Rapid lung alveolarization occurs in the human infant from 36 weeks of gestation through early childhood, while in the mouse, this occurs from 5 to 20 days post birth [42,43]. Our 3-day-old murine neonatal model recapitulates the lung function and immune responsiveness of a late preterm human infant, the group at highest risk of developing severe respiratory viral disease [31]. These models allow for more relevant characterization of age-specific responses to infectious agents, which are driven by multiple biological and developmental factors.

Using our neonatal murine model, we have shown that the immune system of IV-infected neonates is transcriptionally quiescent during the first 12 h post infection (hpi). The targeted analysis of 753 immune-related genes via Nanostring technology revealed similar expression profiles between neonatal lungs at 12 hpi and uninfected controls [44]. This contrasts with observations in adult mice, which exhibit strong expression of immune-related genes at 12 hpi. At 24 hpi, IV-infected neonates begin to regulate gene expression; however, expression is still distinct from adults, highlighting age-related differences in the immune response to IV. A similar dampened host response has been observed in neonatal sepsis. Wynn et al. compared whole-blood gene expression based on age. They found that neonates with septic shock upregulated the fewest number of genes and had reduced expression of pathogen recognition pathways relative to infants, toddlers, and school-aged children [45]. When taken together, these findings suggest neonates exhibit a defect in the ability to initiate immune responses against pathogens.

In neonates, tolerance drives a Th2-skewed anti-inflammatory immune response to mitigate host damage; however, this response delays immune activation to pathogens [36,37]. Therefore, prophylactic mechanisms to improve neonatal immune responses have been investigated. Our group has tested the efficacy of prophylactic treatment with *Lactobacillus rhamonus* (LGG), a Gram-positive commensal bacterium [44]. We observed increased survival coupled with decreased viral load early in infection. Importantly, LGG pre-treatment increased transcription of immune-related genes at 12 hpi in IV-infected neonates, including those related to IFN signaling. However, LGG-mediated protection was significantly reduced with therapeutic application *following* IV infection. Recent work has supported the efficacy of engineered *E. coli* to protect adult mice against lethal IV challenge [46], while Valentin et al. showed that the maternal administration of LGG improved IV clearance in neonatal murine offspring [47]. Similarly to findings in the LGG-treated neonates, probiotic administration increased IFN levels in the lung. However, the efficacy of the prophylactic treatment in IV-infected neonates, but not post infection, suggests that the *timing* of the IFN response is key.

### 2.2. Timing of IFN Signaling Impacts Severity of Respiratory Viral Infections

Of the IFN-I subsets, IFN-β has the highest affinity for IFNAR, leading to a potent response [48,49,50]. We tested the timing of IFN-β treatment on neonatal murine outcome to IV infection. We found that IFN-β treatment *prior* to infection improved the survival of IV-infected neonates, while treatment *following* infection accelerated mortality; this underscored the importance of the timing of IFN production to neonatal IV pathogenesis. Similar findings have been observed in adult models of IV infection, where the prophylactic treatment of IFN-α improved survival and reduced morbidity, while post-treatment enhanced mortality [16,19]. Interestingly, both pre- and post-treatment with IFN-III improved survival [16]. Dijkman et al. reported similar findings, where prophylactic and early therapeutic IFN-III supplementation improved survival during MERS-CoV infection; however, they noted that treatment too late in an infection enhanced mortality [51]. These findings highlight the differential impact and timing of IFN-III and IFN-I signaling on viral pathogenesis.

### 2.3. Developmental Regulation of the IFN Response

IFN expression is thought to be developmentally regulated, with infants having lower expression at birth that then increases over the first year of life [52,53]. Consistent with this, a longitudinal study conducted by Holt et al. reported a developmental deficiency in IFN gene expression from infancy to childhood [54]. One-fourth of CBMCs did not express IFN-Is or IFN-IIIs following stimulation with poly I:C; however, gene expression was observed in PBMCs collected at mid- and late-childhood. Notably, the lack of IFN expression in CBMCs correlated with an increased risk for the development of febrile lower respiratory infections. Similarly, Thwaites et al. found that decreased IFN-I expression in infant nasopharyngeal aspirates correlated with severe RSV-associated bronchiolitis, while Taveras et al. correlated increased IFN-III concentrations in pediatric nasal swabs with less severe RSV outpatient cases [55,56]. Conversely, Selvaggi et al. demonstrated increased IFN-III transcription in RSV-infected neonates positively correlated with an increased respiration rate, a sign of enhanced morbidity [57]. Together, these studies demonstrate how IFNs potentially play a dual role in neonatal respiratory viral pathogenesis.

In adult models of IV infection, IFN-IIIs are the predominant IFNs produced in the airways and exhibit sustained production and antiviral gene expression throughout the first week of infection [17,21,25,26,58]. In contrast, IFN-Is are potent activators of ISGs but its expression declines more rapidly [17,18,21,58]. However, dysregulated IFN-I secretion can drive inflammation and contribute to viral pathogenesis [59]. Diminished IFN signatures are associated with increased disease severity in adults with SARS-CoV-2 infections [60,61]. This dysregulated IFN expression in the adult is thought to impede the initiation of adaptive immunity and instead continues to drive pro-inflammatory cytokine and chemokine release [62,63]. Contrasting findings in adults, recent work has shown that clinical SARS-CoV-2 infection in neonates correlates with a robust IFN response, highlighting the age-specific regulation of IFNs in respiratory viral infections [64].

## 3. Potential Players in IFN-Mediated Viral Pathogenesis in Neonates

We have recently shown that dysregulated IFN-I signaling in murine neonates drives pathogenesis independent of viral load, and its disruption significantly improves survival [23]. The relative contribution of IFN-Is and IFN-IIIs to the protection or susceptibility of neonates to respiratory viral infection requires more study. In neonates, aberrant IFN production may exacerbate an ongoing immune response, leading to excess inflammation, lung damage, and ultimately mortality. Possible mechanisms contributing to IFN toxicity in IV-infected neonates are discussed.

### 3.1. Inhibition of Lung Development

Viral control is one of the main functions of IFNs. As a result, IFNs drive an antiproliferative environment in the lung, which prevents viral spread [65,66]. We suggest that in adults with fully developed lungs, the IFN-induced lapse in lung repair is better tolerated. However, in neonates with rapidly developing lungs, this drives pathology and enhanced morbidity. Recently, Major et al. showed that IFN-I and IFN-III supplementation following IV infection significantly reduced the frequency of proliferating murine type II alveolar epithelial cells (TIIECs) via the induction of cell death and p53-directed antiproliferative effects [21]. A blockade of this signaling using transgenic mice-improved cell proliferation, with IFN-III ablation improving cell proliferation more than IFN-I ablation, suggests that IFN-IIIs primarily impede cell repair following infection. Our recent data suggests that in neonates, IFN-Is are the primary IFN blocking lung repair. We reported that IFN-I signaling in murine neonatal TIIECs downregulated genes associated with Wnt signaling at 2 days post infection, and continued downregulation within the whole lung at 6 days post infection, coinciding with the onset of mortality [67].

TIIECs only comprise 5% of the alveoli; however, they perform key functions for lung development and homeostasis [68,69]. Importantly, TIIECs are the main producers of surfactants in the lungs. These are lipid proteins that facilitate breathing and modulate immune responses to pathogens [70,71]. Furthermore, TIIEC differentiation drives the development of distal alveoli, which form the delicate air–liquid interface required for gas exchange [42]. TIIECs are considered “stem-cell-like” because they have the capacity to self-renew. However, they also differentiate into type I alveolar epithelial cells (TIECs), which make up ~95% of the alveolar epithelium and are crucial for gas exchange. TIIECs are the main target of IV infections, and loss of TIIECs has been shown to negatively impact lung repair in cases of severe damage and respiratory distress [72,73,74].

Wnt/β-catenin signaling plays a role in lung development and regeneration by regulating the self-renewal capacity of TIIECs [75,76,77,78]. A blockade of this signaling has been associated with reduced stemness and increased differentiation into TIECs [75]. Maintenance of stemness is important for lung repair following injury [79,80]. However, active Wnt signaling during IV infection has been shown to increase IV mRNA expression in murine epithelial cell lines [81]. Studies have found that β-catenin, the central component of Wnt signaling, is suppressed by IFN-Is and IFN-IIIs, which reduces cell proliferation and promotes apoptosis in cell lines [82]. Similarly, Hancock et al. showed that infected primary murine TIIECs exhibited increased IFN expression coupled with decreased Wnt signaling relative to uninfected bystander cells [83]. This suggests that Wnt signaling modulation by IFNs serves to control viral replication. However, in the neonate, where survival is tightly linked to lung development, this enhances pathology.

### 3.2. Dysregulation of Chemical Barrier Defenses

Antimicrobial peptides (AMPs) such as defensins, lactoferrins, and cathelicidins are secreted by the pulmonary epithelium and resident immune cells to aid in pathogen clearance, cell growth, and tissue development [84,85]. AMP expression is thought to be developmentally regulated [86]. Lower levels of circulating AMPs have been observed in neonatal blood, and impaired release was found at the site of infections [87,88,89]. AMP expression leads to cell death, immune activation, and compromised lung repair [90,91,92,93]. Recently, we illustrated a pathogenic effect of Cathelicidin-related antimicrobial peptide (CRAMP), the murine analog to human LL-37, during neonatal IV infection [94]. Ablating CRAMP expression improved neonatal survival but had no effect on adult morbidity, highlighting the age-specific pathogenic role of CRAMP during neonatal infection. Zhang et al. reported that LL-37 drives IFN-I production from keratinocytes during skin injury [95]. While IFN-I secretion led to immune cell maturation, it also enhanced inflammation. We noted that IV-infected neonates significantly increased pro-inflammatory cytokine expression in the presence of CRAMP late in infection, coinciding with the onset of mortality and lung damage [94].

### 3.3. Disruption of the Alveolar Barrier Integrity

Within the delicate alveoli, barrier integrity is paramount in preventing excessive fluid influx into the alveolar space, a hallmark of acute respiratory distress syndrome [96,97,98,99]. The lung epithelium forms a physical and immunochemical barrier between the external and internal environments, serving as the first line of defense against inhaled pathogens. Tight junctions (TJs) and adherens junctions (AJs) help form the physical barrier, and their expression and function are developmentally regulated [100]. During early gestation, the pulmonary expression of claudin-5 was found to be increased in alveolar epithelial cells [101]. Claudin-5 expression positively correlates with barrier permeability [102]. At birth, this regulates the transition from a liquid–liquid (LLI) to an air–liquid interface (ALI) [103]. The modulation of the epithelial barrier is critical for the adaptation to an oxygen-rich environment. However, this developmentally necessary process can also increase susceptibility to lung injury during systemic infections in the developing lungs.

TJs and AJs form physical connections between neighboring cells in the alveoli, and their disruption is associated with enhanced lung damage during viral infections [104,105,106]. TJs are formed by interactions between transmembrane (occludins and claudins) and scaffolding (zonula occludens, ZO) proteins [97]. Respiratory viruses downregulate the expression of key TJ proteins, including ZO-1, occludin, and claudin-1,-2,-4 [96,106,107]. Some viruses, like adenovirus, directly bind to TJ proteins on epithelial cells for viral entry, in turn disrupting barrier integrity [97]. Vaswani et al. reported that microRNA-193b-5p induced by IV infection directly binds to the 3′ untranslated region of the occludin gene, disrupting its expression [108]. A blockade of microRNA-193b-5p restores occludin expression, decreases viral replication, and improves survival in adult mice infected with IV. Of note, microRNA-193b-5p is an ISG and its expression is increased by IFN-β.

IFN signaling has been shown to modulate the expression of TJ proteins, including claudin-4 [109]. Claudin-4 regulates alveolar fluid clearance and is often upregulated in lung injury, suggesting it plays an important role in lung homeostasis and repair [110,111]. A follow-up study found that claudin-4 contributed to IFN-driven antiproliferative activity in cells [112]. Our group reported a decrease in cell repair pathways coupled with reduced claudin-8 expression downstream of IFN-I signaling in neonatal TIIECs [67]. Late in the infection we observed the exacerbated loss of claudin-8 in neonates with functional IFN-I signaling. This corresponded to enhanced lung damage relative to neonates lacking IFN-I signaling [23]. Lokken-Toyli et al. reported a decrease in claudin-8 in early life corresponding with increased barrier permeability [113]. This was found to drive the systemic dissemination of *Streptococcus pneumoniae*, a bacterial pathogen which causes 300,000 deaths in children under the age of five, annually [113,114]. Importantly, claudin-8 is an occludin co-recruiting molecule [115]. Occludin is a key sealing junction protein in the lung decreased by IV infection [108,116,117]. Consistent with this, we observed a significant reduction in whole lung occludin expression in neonates with IFN-I signaling, which coincided with worse pathology severity scores [23].

### 3.4. Exacerbation of Oxidative Stress

Reactive oxygen species (ROS) and inflammatory cytokines, such as TNF-α, are associated with pulmonary TJ disruption and enhanced lung pathology [97,118,119]. Virally induced ROS has been shown to enhance IV replication but also aids in neutrophil activation and viral clearance [120,121,122,123]. Of note, the AMP LL-37 is known to promote ROS production from neutrophils and macrophages [124,125]. Similarly, IFNs contribute to early immune cell activation and increase ROS expression from neutrophils and macrophages [126,127]. During viral hepatitis, IFN-I signaling was found to increase ROS production in the liver, and an IFNAR blockade was found to reduce liver damage [128]. We reported an IFN-dependent increase in ROS production from IV-infected neonatal TIIECs, which was ablated with the disruption of IFN-I signaling [23]. Genetic ablation or the targeted inhibition of ROS reduces morbidity and promotes infection resolution during IV infection [129,130].

Excessive or prolonged production of ROS leads to the development of oxidative stress (OS). OS occurs when ROS outweighs the ability for antioxidants to metabolize these free radicals [131]. At baseline, neonates are highly susceptible to developing pulmonary OS due to developmental deficiencies in antioxidant production [132,133]. In humans, neonates admitted to the pediatric intensive care unit for RSV infections had significantly reduced expression of Nrf2, a key regulator of the cellular antioxidant response, compared to children with less severe diseases [134]. In primary human TIIECs, Nrf2 knockdown during IV infection sensitized cells to death, while overexpression decreased viral replication and OS [135]. We have recently reported an age-dependent propensity of IFN-Is in the development of pulmonary OS [23]. IV infection increases pulmonary ROS, which we suggest overloads the already compromised antioxidant system [136,137,138].

Antioxidant expression increases during the third trimester to balance ROS induced by transitioning to an oxygen-rich environment [139]. Maternal transfer of non-enzymatic antioxidants, such as vitamin A, also occurs during this time. Vitamin A signaling is a well-recognized regulator of TIIEC stemness and lung development [140]. In fact, vitamin A supplementation is provided to preterm infants to support lung development and prevent bronchopulmonary dysplasia [141]. Vitamin A deficiency has been linked to impaired alveolar morphogenesis and TIIEC surfactant production [142]. The overexpression of antioxidants decreased lung injury and preserved epithelial cell proliferation and alveolar development in neonatal models of hyperoxia [143,144]. We recently showed that IFN-I signaling decreased the expression of key genes related to vitamin A signaling in IV-infected neonatal TIIECs [67]. This reduction was observed in the whole lung late in infection, corresponding to mortality onset and TJ disruption. Furthermore, the vitamin A derivative, retinoic acid, has been found to modulate TJ properties in the developing lung [103]. Vitamin A also plays a role in the innate immune response to viral infections, and IV-infected mice with a genetic deficiency in vitamin A exhibit increased viral loads, inflammatory cytokine expression, and neutrophil recruitment in the lung [145].

### 3.5. Modulation of Neutrophil Activity

We have previously shown that IFN-I signaling increased inflammatory immune cell recruitment to the lung during neonatal IV infection [23]. Notably, we observed a significant increase in neutrophil infiltration. Neutrophils are the first cells recruited to the site of infection to aid in viral clearance; however, neutrophils are also associated with severe respiratory viral infections [146,147,148]. Neutrophils are a major contributor to ROS and OS during respiratory viral infections [119]. IFN-I signaling increases neutrophil ROS production in tumors, autoimmune diseases, and viral infections [126,149,150]. Furthermore, IFN-I stimulation, but not IFN-III, has been found to increase the transcription of inflammatory cytokines in murine neutrophils [17]. Therefore, IFN-III signaling has been investigated for its relative contribution to the OS response due to its reduced potency. Murine studies report that IFN-IIIs modulate pathogenic neutrophil activation in intestinal, fungal, and viral infections, potentially through the reduction in NADPH signaling [29,151,152]. Neonates lacking IFN-III signaling exhibit neutrophilia, enhanced ROS production, and increased lung damage during pneumovirus infection [29]. Of note, the primary purpose of NADPH oxidase signaling is the production of ROS [153]. The ablation or inhibition of Nox2, the NADPH oxidase expressed in neutrophils and monocytes, decreases cell infiltrate, viral titers, and lung pathology in IV-infected mice [130,154].

Antiviral responses require a delicate balance of sufficient immune activation for viral control without inducing immunopathology. In neonates with underdeveloped lungs and a tolerant immune system, the margin of error is very narrow. Both IFN-Is and OS impair lung repair, disrupt the epithelial barrier, and promote neutrophil activation. More work is needed to elucidate if the enhanced pathogenesis observed in neonates is driven directly by IV, virally induced IFN-Is and ROS, IFN-I-induced ROS, or a combined synergy (Figure 2). Furthermore, investigating the relative contribution of IFN-III signaling will provide a more comprehensive understanding of the role of IFNs in neonatal IV pathogenesis.

## 4. Therapeutic Avenues

Developing safe and effective therapeutics that mitigate disease severity in the infant population is of the highest importance. Understanding the age-specific mechanisms driving increased susceptibility and morbidity to respiratory viral infections is key. Age-appropriate animal models better inform therapeutic targets and potential. IFN-Is and IFN-IIIs are typically viewed as protective innate immune responses against viral infections. However, the pathogenicity of these cytokines is now being recognized. Therapeutics targeting IFN-driven responses associated with severe disease may be effective in this high-risk age group.

IFN modulating therapies are currently FDA approved for the treatment of cancer, viral infections, and autoimmune diseases [155,156]. IFN-I supplementation has been investigated as a therapeutic to mitigate pediatric respiratory disease severity. Treatment with IFN-α is considered well tolerated; however, efficacy is uncertain [157,158,159]. While IFN-α treatment was found to improve symptoms in neonates with RSV pneumonia, the hospitalization stay was not significantly reduced [160]. Of note, IV infection induces a more robust IFN response than RSV, suggesting that IFN supplementation may not be a one-size-fits-all for respiratory viral infections [161]. We propose that the timing of IFN-I treatment is important for efficacy in IV-infected neonates, and supplementation too late in infection will have no effect or even exacerbate the inflammatory response. Therefore, IFN-I blockades may mitigate disease severity in the acute phase of IV infection. This therapy is currently approved for the treatment of systemic lupus erythematosus [162]. In this disease, IFN-α blockades reduce systemic inflammation and symptom severity; however, increased susceptibility to viral infections was noted as a side effect [163].

Recently, IFN-III treatment has been investigated to mitigate severe viral infections [16,51,164,165]. Recent work by Reis et al. reported that IFN-III administration to adults with acute SARS-CoV-2 infections reduced the incidence of hospitalization or emergency visits relative to those who received the control treatment [166]. Of note, these adults were predominantly vaccinated, which is not an option for children under the age of 6 months infected with IV. IFN-III treatment potentially promotes antiviral responses without inducing the excess inflammation observed with IFN-I treatment [16,167]. Recent work by Feld et al. supports this hypothesis; IFN-III administration in adults infected with SARS-CoV-2 enhanced viral clearance with minimal side effects [168]. Therefore, IFN-IIIs may be a viable therapeutic option in the neonatal population. However, timing, treatment dose, and duration need to be well-studied for use in this vulnerable population. Blocking the downstream targets of IFN signaling, such as ROS production, could also be beneficial.

Oxidative stress is linked to multiple organ pathologies, including the development of acute respiratory distress syndrome [169]. We reported that IFN-Is drove severe IV pathology in murine neonates in part due to OS [23]. The mitigation of OS may be a better alternative than the direct modulation of IFNs for neonatal diseases. The administration of the antioxidant N-acetyl cysteine (NAC) has been investigated as a therapy to reduce respiratory viral infection severity. In vitro NAC treatment reduces cell death, pro-inflammatory cytokine expression, and viral replication in A459 cells infected with IV [170,171]. Zhang et al. reported that in vivo NAC treatment during early H9N2 IV infection reduced mortality, inflammatory cytokine production, and pulmonary edema in adult mice [172]. Other studies have tested NAC therapy in combination with antivirals and found that combination therapy significantly improved murine survival following lethal IV infection compared to individual treatments [173,174]. A recent study demonstrated that NAC given prior to delivery in mothers with symptoms of chorioamnionitis, an infection of the placenta and amniotic fluid during pregnancy, improved neurodevelopmental outcomes in the babies [175], which demonstrates how early intervention could improve outcomes. We have previously tested the efficacy of NAC administration in our neonatal murine model of IV infection and found that treatment during acute infection improved survival [23]. However, immunogenicity, poor cell penetration, and molecular half-life need to be optimized to improve therapeutic potential [176].

## 5. Concluding Remarks

Recent work has highlighted the differential roles of IFN-Is and IFN-IIIs during respiratory viral infections. IFN-III signaling is less systemic and promotes localized antiviral immunity without the excess inflammation observed with IFN-I signaling. During IV infection, IFNs impair cell proliferation, disrupt pulmonary barrier integrity, and generate reactive oxygen species. In adults, these effects are outweighed by the antiviral protection IFNs confer. However, age-driven physiological and immunological deficiencies in the neonate instead skew IFNs toward pathogenesis. We have shown that the blockade of IFN-I signaling improves neonatal survival following IV infection, while the loss of IFN-III signaling exacerbates mortality. Proper lung development is mandatory, and the brief lapse in development to control viral replication during infection comes at a significant cost. In addition, neonates possess impaired antioxidant defenses, which induce oxidative stress, leading to cell damage. More work is needed to understand the relative contribution of IFN-Is and IFN-IIIs to cell repair, barrier disruption, and oxidative stress in IV-infected neonates. Insight into the mechanisms driving these effects and their interplay in the neonatal immune response to respiratory viral infections will better inform therapeutic development.

## Figures and Tables

**Figure 1 viruses-17-01002-f001:**
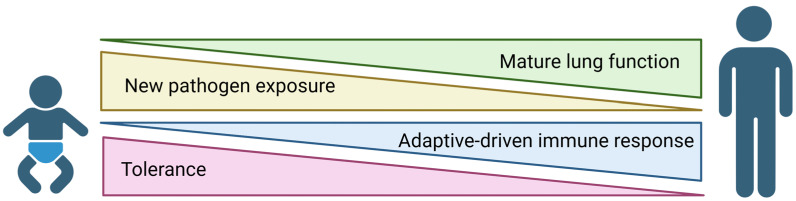
Drivers of age-specific responses to respiratory pathogens. At birth, neonates are constantly exposed to new pathogens; however, their immune system is highly tolerant to prevent host damage [30,36]. As a result, neonates possess underdeveloped adaptive immune responses relative to adults [32,37].

**Figure 2 viruses-17-01002-f002:**
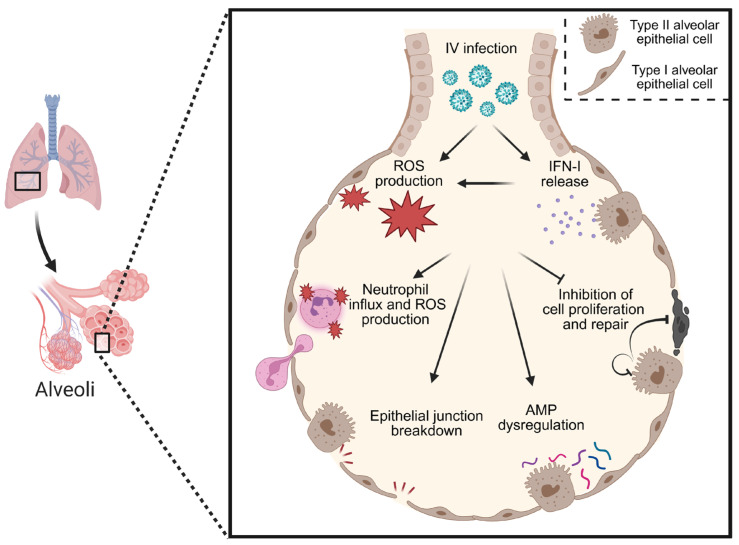
The contribution of IV-induced IFN-I and ROS to viral pathogenesis. Upon infection, IFN-Is are released and ROS production is induced in the lung. IFN-I signaling also contributes to ROS production from epithelial cells. Both IFN-I signaling and ROS production are known to impair cell proliferation and repair, dysregulate AMP production in the lung, disrupt epithelial TJs, and enhance neutrophil recruitment and activation. Understanding the relative contribution of virally induced ROS and IFN-I-induced ROS to these effects is important in elucidating age-specific drivers of severe viral infection.

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
