# Peer review of "Age Matters: Key Contributors to Interferon Toxicity in Infants During Influenza Virus Infection"

_viruses, 2025, doi:10.3390/v17071002_

Round 1
Reviewer 1 Report
Comments and Suggestions for Authors
Influenza virus (IV) infections in premature infants are one of the risk factors for severe diseases, which are accompanied by hospitalization, intubation, development of toxicity and pathogenetic changes, and often lead to a high mortality rate in infants. Therefore, the relevance of the problem discussed in this review is beyond doubt. It is known that interferons (IFNs) are a key component of the innate immune response in the early stages of infection. Currently, there is a discussion in the scientific and clinical literature about the contradictory role of IFNs in protecting the body of newborns from respiratory viral infections, IFN-induced of inflammatory activity and toxicity. In this review, the authors analyze the potential mechanisms of toxicity associated with the production of IFNs in response to IV infection. Prospects for the development of therapeutic agents to counteract the negative effects of IFNs in infants are discussed. The authors have developed a clinically relevant neonatal mouse model of intravenous infection that allows for a more precise characterization of age-related responses to infectious agents. The data obtained in these models are discussed in relation to FDA-approved IFN-modulating therapies for cancer, viral infections, and autoimmune diseases.
The review is written in an understandable language and provides comprehensive evidence for the importance of early administration of interferon therapy for efficacy in IV-infected neonates. This is necessary to take advantage of the narrow window of IFN therapeutic effect. Late administration of interferon therapy during infection will have no effect or will increase the inflammatory response.
In addition, the authors conclude that IFN-I is the main driver of pathogenesis, while IFN-III plays a primarily protective role.
The authors believe that understanding the mechanisms governing these effects and their interactions in the neonatal immune response to respiratory viral infections will allow better prediction of therapeutic developments. The conclusions made by the authors follow logically from the problem under discussion. The cited sources are mostly confirmed by recent publications and are relevant.
The review may be published in the journal Viruses.
Author Response
We appreciate the reviewer’s time and feedback. The reviewer deems the manuscript satisfactory for publication with no additional edits.
Reviewer 2 Report
Comments and Suggestions for Authors
Neonatal immune responses are reliant on innate mechanisms, such as the production of interferon, during the first few months of life. Interferons are important for viral control and clearance. However, recent literature has suggested that interferons may exacerbate disease. This review focuses on the potential mechanisms that drive interferon-induced toxicity in neonates and prospective therapeutics to mitigate this toxicity.
Several suggestions:
- Please check the terms [influenza virus (IV)] [influenza A virus (IAV)] [respiratory viruses (RVs)] again to avoid misunderstanding, e.g., in Fig. 2, [IAV] is used; is it better to use (IV) or (RVs)?
- Please check the terms [interferon (IFN)] [interferon type-I (IFN-I)] [interferon type-III (IFN-III)] again to avoid misunderstanding, e.g., in lines 2 and 19, [interferon] is used; is it better to use [interferon type-I (IFN-I)]; also in fig.2, [IFN-I] is used, is it better to use (IFNs)?
- [Interferons] are a complex issue. Is it possible to explore more regarding the differences between (IFN-I) and (IFN-III) in page 2, such as the activation, signaling pathways, and/or the downstream ISGs?
- Lines 79-80, please add references after [Neonatal immune responses are highly tolerant due to constant exposure to new pathogens].
- Line 168, is it better to add [dysregulated] or other adj. before [IFN-I]?
- Line 267, is there a reference after [worse pathology severity scores]?
- After line 415, abbreviations, [aire-liquid interface] or [air-liquid interface]?
Author Response
We appreciate the reviewer’s consideration and feedback on our manuscript. Below is a detailed response to all comments.
- Please check the terms [influenza virus (IV)] [influenza A virus (IAV)] [respiratory viruses (RVs)] again to avoid misunderstanding, e.g., in Fig. 2, [IAV] is used; is it better to use (IV) or (RVs)?
- Figure 2 has been updated to refer to influenza virus as IV and maintain consistency with the manuscript.
- Please check the terms [interferon (IFN)] [interferon type-I (IFN-I)] [interferon type-III (IFN-III)] again to avoid misunderstanding, e.g., in lines 2 and 19, [interferon] is used; is it better to use [interferon type-I (IFN-I)]; also in fig.2, [IFN-I] is used, is it better to use (IFNs)?
- The manuscript has been updated to use IFN-I to refer to type I interferon, IFN-III for type III interferon, and IFN(s) when referring to interferons as a whole. The legend of Figure 2 (lines 353-359) and lines 346-350 of the manuscript have been adjusted to refer to only IFN-I during IV infection.
- [Interferons] are a complex issue. Is it possible to explore more regarding the differences between (IFN-I) and (IFN-III) in page 2, such as the activation, signaling pathways, and/or the downstream ISGs?
- A paragraph highlighting the differences between IFN-I and IFN-III dependent downstream ISG expression and kinetics has been added (lines 46-60).
- Lines 79-80, please add references after [Neonatal immune responses are highly tolerant due to constant exposure to new pathogens]
- The figure legend has been edited and sources have been added (lines 98-101).
- Line 168, is it better to add [dysregulated] or other adj. before [IFN-I]?
- Dysregulated has been added before IFN-I (line 184).
- Line 267, is there a reference after [worse pathology severity scores]?
- A reference has been added following this sentence (line 284).
- After line 415, abbreviations, [aire-liquid interface] or [air-liquid interface]?
- ALI refers to air-liquid interface, the abbreviation has been updated (after line 438).